# Vericiguat suppresses ventricular tachyarrhythmias inducibility in a rabbit myocardial infarction model

Po-Cheng Chang [1,2], Hui-Ling Lee [2,3], Hung-Ta Wo [1,2], Hao-Tien Liu [1,2], Ming-Shien Wen [1,2], Chung-Chuan Chou [1,2] *

1 Department of Internal Medicine, Division of Cardiology, Chang Gung Memorial Hospital, Linkou, Taoyuan, Taiwan, 2 Medical School, Chang Gung University, Taoyuan, Taiwan, 3 Department of Anesthesia, Chang Gung Memorial Hospital, Taipei, Taiwan

* 2867@adm.cgmh.org.tw

## Abstract

### Background

The VICTORIA trial demonstrated a significant decrease in cardiovascular events through vericiguat therapy. This study aimed to assess the potential mechanisms responsible for the reduction of cardiovascular events with vericiguat therapy in a rabbit model of myocardial infarction (MI).

### Methods

A chronic MI rabbit model was created through coronary artery ligation. Following 4 weeks, the hearts were harvested and Langendorff perfused. Subsequently, electrophysiological examinations and dual voltage-calcium optical mapping studies were conducted at baseline and after administration of vericiguat at a dose of 5 μmol/L.

### Results

Acute vericiguat therapy demonstrated a significant reduction in premature ventricular beat burden and effectively suppressed ventricular arrhythmic inducibility. The electrophysiological influences of vericiguat therapy included an increased ventricular effective refractory period, prolonged action potential duration, and accelerated intracellular calcium ($Ca_i$) homeostasis, leading to the suppression of action potential and $Ca_i$ alternans. The pacing-induced ventricular arrhythmias exhibited a reentrant pattern, attributed to fixed or functional conduction block in the peri-infarct zone. Vericiguat therapy effectively mitigated the formation of cardiac alternans as well as the development of reentrant impulses, providing additional anti-arrhythmic benefits.

### Conclusions

In the MI rabbit model, vericiguat therapy demonstrates anti-ventricular arrhythmia effects. The vericiguat therapy reduces ventricular ectopic beats, inhibiting the initiation of

**Data Availability Statement:** The data is in the Supporting information File.

**Funding:** This study was supported by the Ministry of Science and Technology, Taiwan (110-2314-B-

182A-119- to P.C. Chang) and Chang Gung
Medical Foundation (CMRPG3L1202 to P.C.
Chang). The funders contributed to the study
design, data collection and analysis, and
preparation of the manuscript in this study.

**Competing interests:** The authors have declared
that no competing interests exist.

ventricular arrhythmias. Furthermore, the therapy successfully suppresses cardiac alternans, preventing conduction block and, consequently, the formation of reentry circuits.

## Introduction

Coronary artery disease (CAD) contributed most to the development of heart failure (HF) and is the leading cause of death in developed countries. Sudden cardiac death (SCD) accounts for approximately 50% of all deaths attributed to cardiovascular disease [1]. Ventricular arrhythmia stands as one of the main culprits behind SCD in patients with myocardial infarction (MI). Pharmacological therapies have been shown to reduce mortality in patients with CAD and HF. The advancements in medical therapies for HF with reduced left ventricular ejection fraction (HFrEF) have led to improvements in symptoms and survival, as well as the attenuation of cardiac remodeling and enhancement of heart function. Contemporary pharmacological therapy for HFrEF includes angiotensin-converting enzyme inhibitors, alternative angiotensin II receptor blockers, an angiotensin receptor-neprilysin inhibitor, beta-blockers, aldosterone receptor blockers, and sodium-glucose transport protein 2 inhibitors [2–4]. Recently, a soluble guanylate cyclase (sGC) stimulator, vericiguat, has been proven to be beneficial in patients with HFrEF [5].

Vericiguat, an sGC activator, enhances the production of cGMP, which further stimulates protein kinase G (PKG). The downstream effects of PKG stimulation include vasodilation, reduction of ventricular hypertrophy, and reduction in inflammation as well as cardiac remodeling [6]. HF patients have a relatively high risk of cardiac events within 3 months after hospitalization, and the VICTORIA study enrolled patients with acute HF decompensation [5]. In the VICTORIA study, the vericiguat group experienced an absolute risk reduction of 3.0% in the primary composite outcome (death from cardiovascular causes or hospitalization for HF) as compared with the placebo group, indicating a beneficial effect of vericiguat therapy at the early stage after discharge for HF hospitalization. In this extremely high-risk population (absolute annual death rate ~ 20%), the composite outcome of cardiovascular death and HF hospitalization was reduced 8%. Interestingly, the risk of ventricular tachycardia (VT) was also reduced by 30%. Considering the reduction in cardiovascular deaths observed in this study and the fact that ventricular tachyarrhythmia is the most frequent cause of sudden cardiac death in HF patients [7], we might reasonably attribute a portion of the cardiovascular death reduction to a decrease in cardiac arrhythmias. However, the cardiac electrophysiological mechanisms of vericiguat therapy on cardiac arrhythmias are not yet clear. This study aimed to investigate whether vericiguat stimulates sGC to exert cardiac electrophysiological effects, thereby reducing ventricular arrhythmia events. We performed cardiac electrophysiological examinations and optical mapping to evaluate the effects of vericiguat therapy in ventricular arrhythmia inducibility in a chronic MI rabbit model.

## Methods and materials

### Coronary artery ligation and MI model

The research protocol for this study received approval from the Institutional Animal Care and Use Committee of Chang Gung Memorial Hospital (IACUC approval number: 2022031702) and conformed to the principles outlined in the Guide for Use of Laboratory Animals. New Zealand white rabbits of mixed gender (both male and female), weighing between 2.5 to 4.0 kilograms, were utilized for the experiments. We employed the previously described

techniques to create MI [8]. Prior to the procedure, the rabbits were pre-medicated with intramuscular injections of Zoletil (15 mg/kg) and Rompun (5 mg/kg), and general anesthesia was achieved using isoflurane (1.5–3%) via endotracheal intubation. A left thoracotomy was performed to expose the left ventricle. Ligation of one or two obtuse marginal branches of the left circumflex artery was carried out at the basal one-third region between the atrioventricular groove and the ventricular apex. It is possible that ligation of certain smaller obtuse marginal branches did not induce MI. In some rabbits, a single artery ligation did not cause obvious MI (no cyanotic change) due to collateral circulation, and we ligated another branch to ensure the development of MI. The development of MI was confirmed through observable signs of myocardial ischemia-infarction, including immediate cyanotic changes in the perfused territory, as well as scar formation, observed one month after the surgical procedure. During optical mapping, we visually defined the scar tissue (depicted by the white scar in the left upper panel in S1 Fig), identified it with the fluorescent dye (indicated by the green area in the left lower panel, as healthy tissue emits orange fluorescence), and confirmed it in the optical mapping images (MI area displayed obscure signals, as shown by the signal noise in the right panel). Because the signals obtained from the MI zone are obscure, we excluded the MI zone from the data analyses.

## Electrophysiological study and optical mapping in the rabbit MI model

After 4 weeks, the hearts were harvested following the aforementioned general anesthesia and Langendorff-perfused using oxygenated 37˚C Tyrode's solution. The Tyrode's solution composition, expressed in mmol/L (except as indicated elsewhere), comprises NaCl 125, KCl 4.5, $NaHCO_3$ 24, $NaH_2PO_4$ 1.8, $CaCl_2$ 1.8, $MgCl_2$ 0.5, dextrose 5.5, and bovine serum albumin 100 mg/L, with a pH of 7.40, which is adjusted using 1 mmol/L HCl. The hearts were stained with Rhod-2-AM (5 mmol/L, Molecular Probes, Eugene, OR, USA, dissolved in dimethyl sulfoxide mixed with pluronic F-127) for intracellular $Ca^{2+}$ ($Ca_i$) mapping, and RH-237 (1 mmol/L in 20 mL Tyrode's solution, Molecular Probes, dissolved in dimethyl sulfoxide) was used for membrane potential ($V_m$) imaging. For exciting the fluorescence dyes, we used a laser light source at a wavelength of 532 nm (Millennia, Spectra-Physics Inc., Santa Clara, CA, USA). The emitted fluorescence was filtered (715 nm for $V_m$ and 580 nm for $Ca_i$) and acquired with two MiCAM Ultima cameras (BrainVision, Tokyo, Japan) at a temporal resolution of 2 ms per frame and a spatial resolution of 100 x 100 pixels covering an area of 2.5 cm x 2.5 cm (0.25 x 0.25 $mm^2$ per pixel). In order to mitigate motion artifacts, we utilized a myosin ATPase inhibitor, blebbistatin (10 µmol/L, Tocris Bioscience, Minneapolis, MN), in the Tyrode's solution continuously during the optical mapping recording.

Pseudo-electrocardiography was obtained by placing three electrodes at the left atrium, the posterior wall of the left ventricle, and the right ventricle. Additionally, a bipolar electrode catheter was inserted into the right ventricle through the pulmonary artery for ventricular pacing at twice the threshold. The optical mapping signals were acquired during the electrophysiological study. A ventricular arrhythmia induction protocol, employing both extra-stimulus pacing and burst pacing, was utilized to induce VT and ventricular fibrillation (VF). The extra-stimulus pacing protocol consisted of six S1 beats with an S1-S1 interval of 300 ms, followed by S2-S5 beats starting from an interval of 200 ms, gradually shortened until reaching the ventricular effective refractory period (VERP). The burst pacing protocol started with an S1-S1 interval of 300 ms, which was gradually shortened until a loss of 1 to 1 capturing was observed. To mitigate the effects of cardiac memory, we initiated the acquisition of optical mapping signals after 30 or more paced beats at the same PCL to ensure stable cardiac electrophysiological properties during signal acquisition. In this study, VT was defined as the

occurrence of three or more consecutive organized ventricular beats, while VF was characterized by consecutive disorganized rapid ventricular activities. If VT/VF episodes persisted for 30 seconds or longer, we performed external cardioversion-defibrillation using an external defibrillator (Philips Heartstart XL M4735A, Amsterdam, Netherlands), ranging from 3J to 10J.

Optical mapping data for action potential duration (APD) and $Ca_i$ transient duration ($Ca_iTD$) were obtained with a burst pacing protocol at pacing cycle lengths (PCL) of 200 ms and 300 ms. APD and $Ca_iTD$ were measured at the level of 80% of repolarization. For each heart, APD and $Ca_iTD$ values were obtained from a minimum of 25 non-infarcted sites to calculate the mean APD and $Ca_iTD$. The optical mapping data for the time constant of $Ca_i$ decay were acquired at the termination of a burst pacing protocol at a PCL of 300 ms. The time constant of $Ca_i$ decay (τ value) was measured using monoexponential fitting, with the starting point at 20% repolarization level. APD alternans were defined as a difference of more than 10 ms between consecutive beats, and significant $Ca_i$ alternans were defined as a difference of more than 5% in $Ca_i$ amplitudes between consecutive beats. The analyses of cardiac alternans threshold were based on the analysis of optical mapping data using a burst pacing protocol. Because the magnitudes of alternans were not homogeneous, the sites with maximum alternans were chosen for APD and $Ca_i$ alternans measurement. The optical mapping data were analyzed using LabView 7 (National Instruments, Austin, Texas, USA).

Certain rabbits underwent a repeated electrophysiological study and optical mapping to assess the delayed effects of the Langendorff perfusion on electrophysiological properties. The purpose of these brief delayed tests was to address concerns related to fluorescence dye decay (photobleaching) and alterations of electrophysiological property associated with the excitation-contraction uncoupler [9,10]. We randomly selected 7 rabbits out of the 14 for the delayed electrophysiological tests.

After the baseline experiments, vericiguat at a dose of 5 μmol/L was administrated to evaluate the effects of acute vericiguat therapy. As reported by Cai et al. [11], vericiguat at a dose of 1 to 10 μmol/L is considered safe and capable of promoting an increase in cGMP in cardiomyocytes. The same pacing protocols for electrophysiological study and optical mapping were repeated 20 minutes after the start of vericiguat perfusion.

## Data analysis

Continuous variables with normal distribution were expressed as the mean ± SD, while categorical variables were expressed as numbers (percentage). Continuous variables between two groups were compared using paired Student's t-test. Categorical variables were compared using Fisher's exact test. Statistical significance was considered when the P value was < 0.05.

## Results

### MI creation and heart function

A total of 20 rabbits underwent MI creation. During the MI creation procedure, three rabbits did not survive, and an additional three rabbits experienced sudden death several days after the surgical procedure. We included a total of 7 female and 7 male rabbits in the optical mapping analyses. The remaining 14 rabbits received cardiac electrophysiological study and optical mapping examinations. The average age of rabbits during the optical mapping study was 25.1 ± 2.5 weeks. After 4 weeks, cardiac echocardiography revealed a mean left ventricular ejection fraction (LVEF) of 38.3 ± 9.2%.

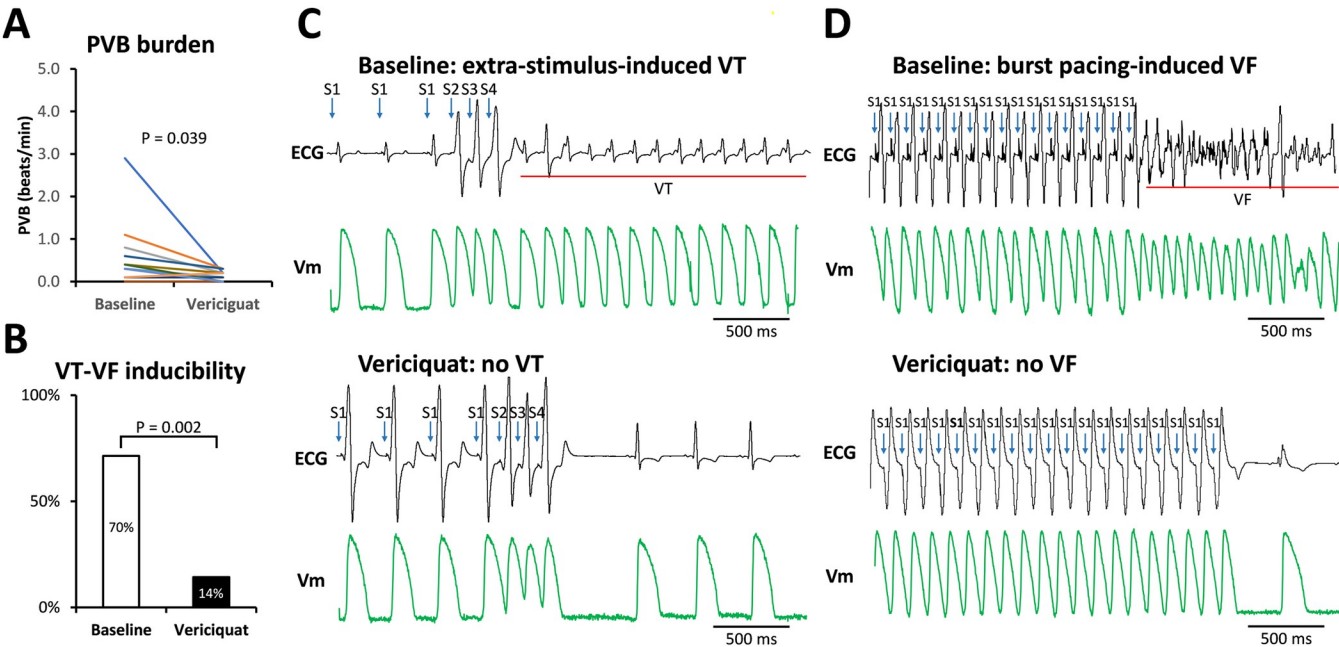

**Fig 1. Post-MI PVB burden and VT/VF inducibility (N = 14). A.** Summarized results of the PVB burden: Vericiguat therapy significantly reduced the PVB burden. **B.** Summarized results of the VT/VF inducibility in the electrophysiological exams. Vericiguat therapy demonstrated significant reduction in VT/VF inducibility. **C.** Representative examples of extrastimulus-induced ventricular arrhythmias at baseline and non-inducibility after vericiguat therapy. **D.** Representative examples of burst pacing-induced ventricular arrhythmias at baseline and non-inducibility after vericiguat therapy. The comparisons of the representative traces were acquired from the same rabbit. MI, myocardial infarction; PVB, premature ventricular beat; VF, ventricular fibrillation; $V_m$, membrane potential; VT, ventricular tachycardia.

## Ventricular arrhythmia inducibility

We first explored the burden of spontaneous ventricular premature beats (PVBs). At baseline, the PVB burden was 0.55 ± 0.74 beats per minute. After vericiguat infusion, there was a significant reduction in spontaneous PVBs (0.12 ± 0.11 beat/min, P = 0.039, Fig 1A). The cardiac electrophysiological study demonstrated a significant reduction in ventricular arrhythmia (VT/VF) inducibility (70% before vs 14% after vericiguat infusion, P = 0.002, Fig 1B). Fig 1C and 1D show representative examples of ECG and action potential (membrane potential, $V_m$) during extrastimulus-induced VT and VF.

## Effects of vericiguat therapy on electrophysiological properties

We further examined the effects of acute vericiguat therapy on the electrophysiological mechanisms involved in suppressing ventricular arrhythmia inducibility. We performed optical mapping to evaluate the electrophysiological properties both at baseline and after acute vericiguat therapy. Acute vericiguat therapy resulted in prolongation of the VERP (136.4 ± 8.4 ms to 149.2 ± 8.6 ms, P < 0.001, Fig 2A). Additionally, acute vericiguat therapy increased APD at a PCL of 300 ms (Fig 2B, 123.1 ± 8.5 ms to 132.6 ± 5.5 ms, P < 0.001) and a PCL of 200 ms (Fig 2C, 107.0 ± 5.4 ms to 117.2 ± 6.1 ms, P < 0.001). Although VERP is correlated with APD, the extent of VERP prolongation was greater than the extent of APD increase. Acute vericiguat therapy did not change $Ca_iTD$ at a PCL of 300 ms (Fig 2D, 142.3 ± 9.6 ms vs. 142.4 ± 9.4 ms, P = 0.938) and a PCL of 200 ms (Fig 2E, 119.8 ± 4.2 ms vs. 122.3 ± 4.1 ms, P = 0.068). The bottom subpanels display representative ECG, $V_m$, and $Ca_i$ traces at baseline and after vericiguat therapy. The thresholds of burst pacing-induced alternans were significantly shortened after

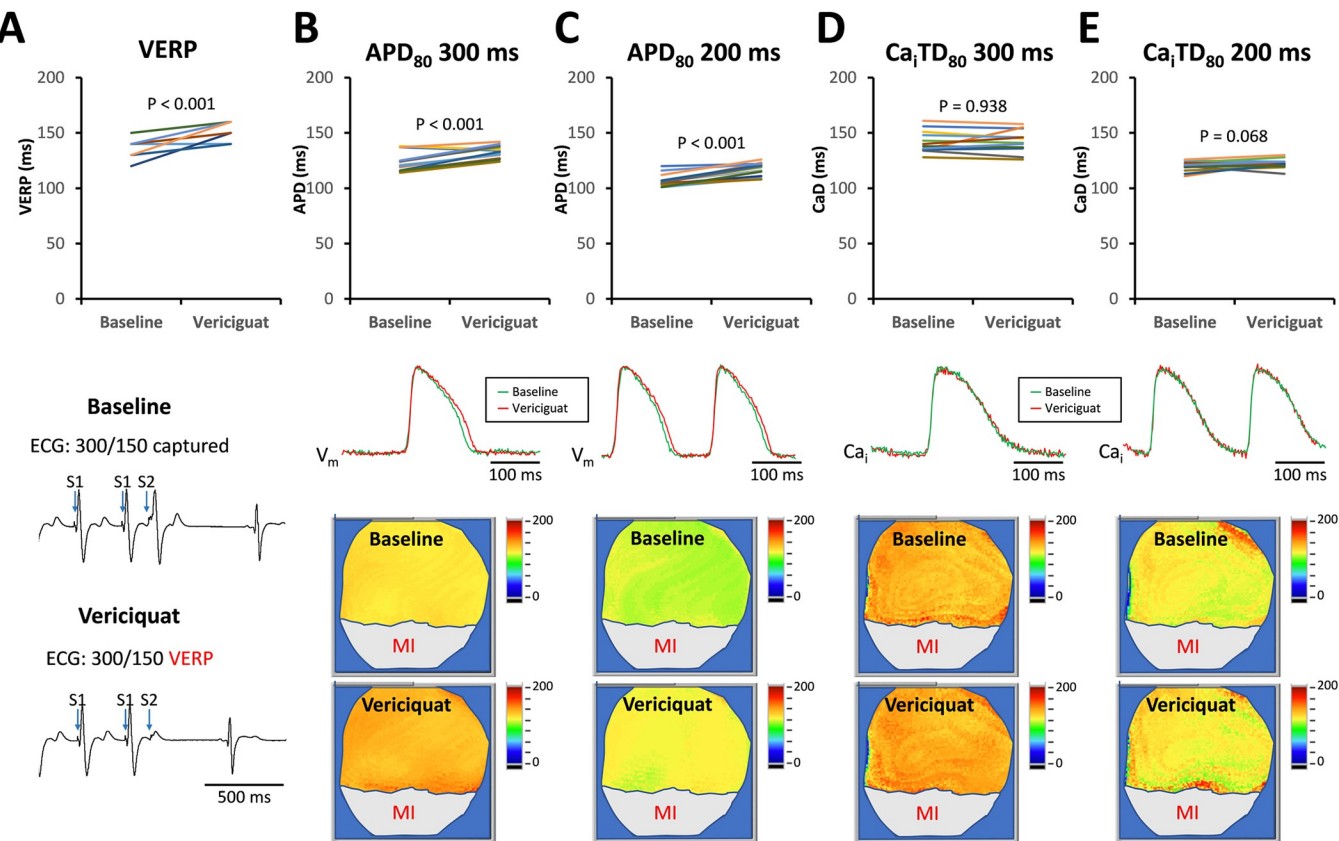

**Fig 2. Effects of vericiguat therapy on VERP, APD$_{80}$, and CaD$_{80}$ (N = 14). A.** VERP and the representative ECG traces. **B.** APD$_{80}$ at pacing cycle length 300 ms and the representative AP traces. **C.** APD$_{80}$ at pacing cycle length 200 ms and the representative AP traces. **D.** CaD$_{80}$ at pacing cycle length 300 ms and the representative Cai traces. **E.** CaD$_{80}$ at pacing cycle length 200 ms and the representative Cai traces. The comparisons of the representative traces and maps were acquired from the same rabbit. APD$_{80}$, action potential duration measured to 80% repolarization; CaD$_{80}$, intracellular calcium transient duration measured to 80% repolarization; VERP, ventricular effective refractory period.

vericiguat therapy (127.1 ± 10.7 ms to 115.0 ± 15.1 ms, P = 0.004, Fig 3A). Fig 3B demonstrates representative examples of low-high alternans of the Ca$_i$ traces and short-low alternans of the V$_m$ at baseline, as well as the V$_m$ and Ca$_i$ traces after vericiguat therapy.

Although vericiguat therapy did not change Ca$_i$TD, it significantly enhanced Ca$_i$ homeostasis. Acute vericiguat therapy significantly shortened the tau value of Ca$_i$ decay (50.3 ± 4.8 ms to 47.8 ± 4.3 ms, P = 0.001). Fig 4A revealed the mean Ca$_i$ decay tau values at baseline and after vericiguat therapy, and Fig 4B shows representative traces of Ca$_i$ and the calculated tau value.

To exclude the possibility of temporal changes in electrophysiological properties, we repeated the electrophysiological tests 20 minutes after the first exams in 7 rabbits. None of the cardiac electrophysiological properties showed significant changes in the delayed exams. After the brief delayed exams, we then infused vericiguat to complete the vericiguat therapy exams. The cardiac electrophysiological properties in the delayed tests are shown in the S2 and S3 Figs.

## The pattern of ventricular arrhythmia in this model

Among the 10 rabbits who developed VT/VF, 9 rabbits (90%) presented a pattern of one or two initial triggered ectopic beats originating from the peri-infarct border areas, followed by reentrant VT/VF. The other rabbit presented with VF of multiple migratory reentrant circuits.

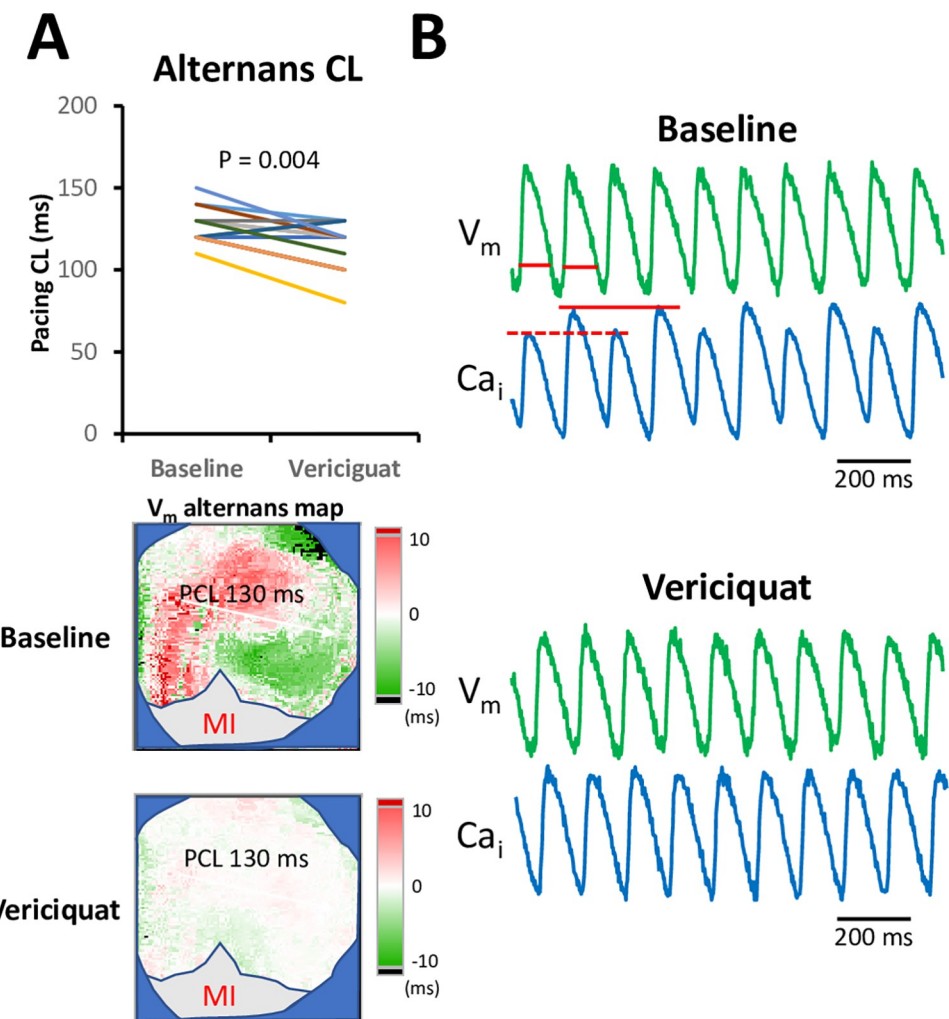

**Fig 3. Effects of vericiguat therapy on alternans (N = 14). A.** Summarized results of the cardiac alternans before and after the acute vericiguat therapy. **B.** Representative $V_m$ and $Ca_i$ traces. The comparisons of the representative traces and maps were acquired from the same rabbit. PCL, pacing cycle length.

Fig 5 shows an example of the pacing-induced ventricular tachycardia pattern. We created anterior and lateral wall myocardial infarction (Panel A), and the rabbit developed HF (LVEF 31.6%). Optical mapping and electrophysiological study demonstrated $V_m$ and $Ca_i$ discordant alternans (Panel B) at a burst PCL of 110 ms. Extrastimulus induced an episode of VT, and Panel C shows the ECG and $V_m$ traces of the episode of pacing-induced VT. The pattern of this VT episode was reentrant around the peri-infarct area. The first VT beat was associated with the conduction block at the nodal line (the blue lines in panel B) of the discordant alternans. Panel D shows the isochrone and phase maps of selected beats in this VT episode, in which the initial S1-paced beats originated from the right ventricle, and the S2-4 beats developed conduction block at the nodal line of discordant alternans. In the phase maps, the VT beat 2 showed a reentrant pattern surrounding a pivot point (see the arrowhead), which colocalized with the peri-infarct zone of the nodal line of discordant alternans.

After vericiguat therapy, both the $V_m$ and $Ca_i$ alternans were suppressed. Fig 5E shows the representative $V_m$ and $Ca_i$ restitution maps during vericiguat infusion in the failing heart. At a burst pacing cycle length of 110 ms, the restitution maps did not reveal significant $V_m$ and $Ca_i$

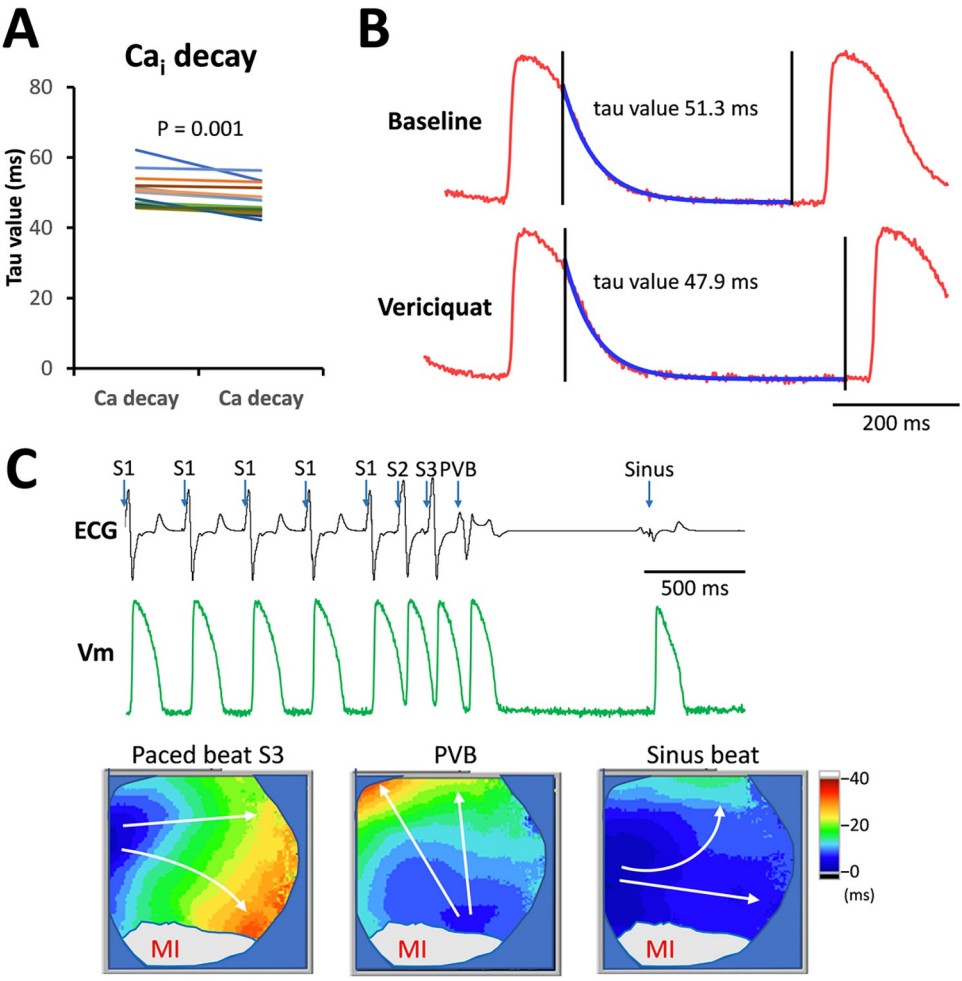

**Fig 4. Calcium decay (tau value) and represenative patterns of a paced beat, a premature ventricular beat, and a sinus beat. A.** Calcium decay (Tau) analyses showed significant shortening of tau value after vericiguat therapy (N = 14). **B.** Representative $Ca_i$ decay traces and tau value analyses at baseline and after vericiguat therapy. **C.** Representative ECG and $V_m$ traces and the patterns of an S3 paced beat, a PVB, and a sinus beat. The comparisons of the representative traces were acquired from the same rabbit.

alternans. Using the same electrophysiological pacing protocol, extra-stimulus pacing did not induce ventricular arrhythmias (Fig 5F). The isochrone maps (Panel G) showed neither functional conduction block nor development of reentrant arrhythmias.

## Discussions

### Major findings

In this study, acute vericiguat therapy exhibited anti-ventricular arrhythmia effects, including the reduction of spontaneous PVB burden and the suppression of ventricular arrhythmia inducibility. The cardiac electrophysiological effects of acute vericiguat therapy included the increase of VERP, prolongation of APD, $Ca_i$ homeostasis, and suppression of $V_m$ and $Ca_i$ alternans. The acceleration of $Ca_i$ homeostasis, represented by the shortening of the $Ca_i$ decay tau value, also contributed to the suppression of Vm and $Ca_i$ alternans. The combined effects of acute vericiguat therapy suppressed $V_m$ and $Ca_i$ alternans to prevent functional conduction block during electrical stimulation. Consequently, we observed that most of the pacing-

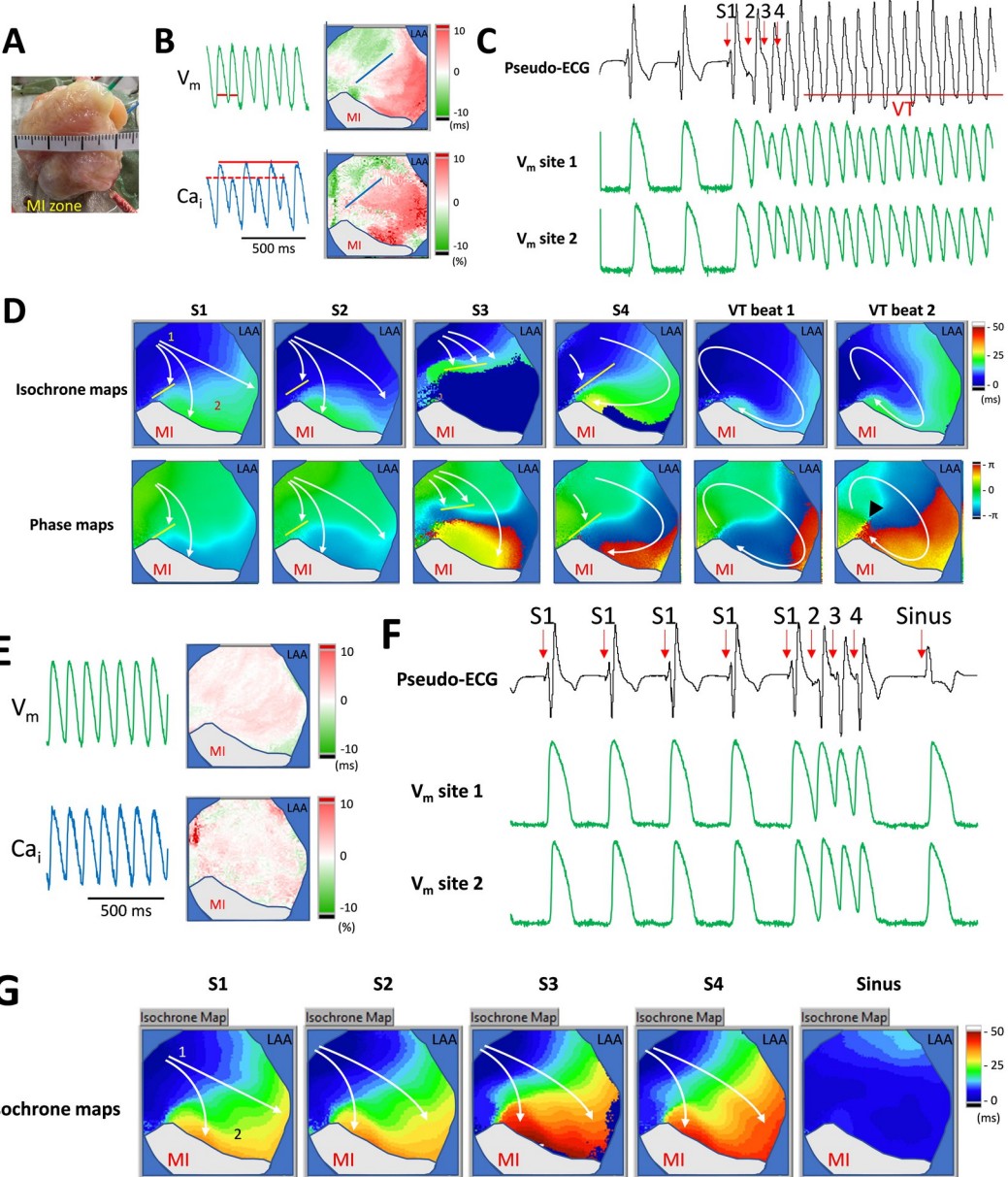

**Fig 5. The pattern of a ventricular arrhythmia in this animal model. A.** A representative Langendorff perfused heart and its schematic diagram of the MI zone. **B.** Representative $V_m$ and $Ca_i$ traces, and the corresponding alternans maps. At a pacing-cycle length of 110 ms, the alternans maps revealed significant discordant alternans. **C**. Pseudo-ECG trace and Vm trace. VT was induced followed by the extra-stimuli. Sites 1 and 2 represent the recorded site of the Vm traces indicated in the S2 isochrone map of Panel D. **D.** Isochrone maps (upper subpanels) and phase maps (lower subpanels) show action potential propagation during an episode of ventricular tachycardia. The S1 to S4 beats had conduction block at the left aspects on the maps near the MI zone. After extra-stimuli, the VT beats (VT1 and VT2) show a reentrant pattern surrounding a phase singularity point (black arrowhead on the phase map of VT2) over the peri-infarct area. White arrows represent the propagation direction of the particular beats.

induced VT/VF exhibited a reentrant pattern, associated with fixed or dynamic functional conduction block at the area of the peri-infarct zone. Acute vericiguat therapy suppressed the formation of alternans as well as the generation of reentrant impulses.

## Mechanisms of ventricular arrhythmia suppression

The exact mechanisms of ventricular arrhythmia suppression through acute vericiguat therapy are not clear. The molecular mechanisms underlying the beneficial effects of acute vericiguat therapy might involve the restoration of reduced nitric oxide (NO) bioavailability and enhanced cGMP production. In failing hearts, endothelial dysfunction leads to reduced NO bioavailability, subsequently causing decreased sGC activity and diminished cGMP production [12]. Vericiguat directly enhances sGC independently of circulating NO and indirectly sensitizes sGC to endogenous NO [13]. Increased cGMP offers benefits to failing hearts by reducing left ventricular afterload through vasorelaxation, decreasing arterial thrombosis, ameliorating apoptosis, and inhibiting remodeling through anti-inflammatory and anti-fibrotic mechanisms. Beyond its effects on ventricular remodeling, activation of the NO-sGC-cGMP signaling pathway may also impact ion-channel remodeling. In another animal study, it was shown that carbamylcholine, an activator of NO, effectively reduced the inducibility of ventricular arrhythmia. The effect was associated with raising the ventricular fibrillation threshold, flattening the action potential restitution curve, and prolonging APD [14]. Interestingly, the study found that the administration of ODQ, an inhibitor of sGC, reversed the increase in VF threshold caused by carbamylcholine. These observations underscore the significant role played by sGC activation in the suppression of NO-associated ventricular arrhythmia.

## Mechanisms of VERP and APD prolongation

The influence of sGC activation on ion channels is complicated. Activation of sGC leads to prolongation of APD and increase of VERP, both of which are pivotal in preventing the emergence of rapid reentrant impulses. However, the precise mechanisms underpinning the prolongation of APD and VERP remain somewhat elusive. In a patch-clamp investigation, it was found that saturated NO solution inhibited the whole-cell $Na^+$ current ($I_{Na}$) in isolated cardiomyocytes [15]. The study also revealed that 8-bromo-cGMP, a cell-permeable cGMP analog, displayed inhibitory effect on $I_{Na}$ [15]. Moreover ODQ, a NO-sensitive sGC inhibitor, partially counteracted the inhibitory effect of NO on $I_{Na}$, thereby indicating a modulatory role for cGMP in $I_{Na}$ regulation. These findings suggest that the NO signaling system exerts inhibitory control over $I_{Na}$ through mechanisms that involve cGMP. The inhibition of $I_{Na}$ by cGMP can elucidate the extension of the VERP.

This study revealed that the activation of sGC led to increase in APD, aligned with findings from a previous study involving a NO activator, carbamylcholine [14]. While the inhibition of the $I_{Na}$ could potentially lead to APD shortening, another study [16] demonstrated that a cGMP-analogue, 8-Br-cGMP, prolonged APD by inhibiting $I_{Kr}$ through the PKG-dependent pathway. Notably, the prolonged APD effects were abolished by an $I_{Kr}$ inhibitor E-4031.

In addition to the effects of sGC stimulation on $I_{Kr}$, NO release might also increase late $I_{Na}$. Caveolin-3 and α1-Syntrophin bind to the $Na^+$ channel, negatively regulating the activity of neural NO synthase (nNOS). In a HEK cell study of long QT type 9, Caveolin-3 mutation causes the loss of inhibitory effect on nNOS, and the enhanced function of nNOS induced accentuated local NO, causing increased late $I_{Na}$ via S-nitrosylation of $Na^+$ channel [17]. A nNOS inhibitor, L-NMMA eliminated the effects of increased late $I_{Na}$ [18]. These studies provide some clues that sGC stimulation might also increase late $I_{Na}$. Consequently, sGC stimulation causes $I_{Kr}$ inhibition, and increased late $I_{Na}$, leading to APD prolongation. However, a study showed that long-term NO donor application did not affect the expression of $I_{Na}$ [19], implying that the regulation of $I_{Na}$ via the NO-sGC-cGMP pathway is far more complicated, and the mechanisms are still unclear. Besides $I_{Kr}$, late $I_{Na}$, and $I_{NCX}$, some other ion channels might also contribute to the regulation of APD and myocardial conduction.

## Suppression of $V_m$ and $Ca_i$ alternans

In this study, we discovered that vericiguat accelerated $Ca_i$ homeostasis, resulting in a shorter $Ca_i$ decay tau value and suppression of $V_m$ and $Ca_i$ alternans. The mechanisms underlying this shortened $Ca_i$ decay can be explained through reduction in inward $I_{Ca}$, decrease in leak of ryanodine receptor (RyR), and improvement of sarcoplasmic reticulum (SR) function. A patch clamp study demonstrated that exogenous applied PKG can inhibit the $I_{Ca}$ stimulatory effects by beta-adrenergic receptor agonist [20]. Moreover, YC-1, a NO-independent sGC activator, reduced beta agonist-stimulated $I_{Ca}$ [21] in another study. These observations collectively suggest a correlation between sGC activation and the attenuation of trans-sarcolemma calcium influx, subsequently mitigating calcium load within SR.

In addition to ion channels on the sarcolemma, the activation of sGC causes phosphorylation of RYR Ser-2808 to modulate RYR leak [22]. Activation of PKG phosphorylates SR regulator protein phospholamban. This phosphorylation enhances $Ca_i$ reuptake by SR $Ca^{2+}$-ATPase (SERCA). Notably, in an sGC knock-out mouse study, the administration of lipopolysaccharide (LPS) induced more pronounced abnormalities in $Ca_i$ dynamics, resulting in diminished change in $Ca_i$ level and prolongation of $\tau$ value in the sGCα1$^{-/-}$ mice than in wild-type mice [23]. The $Ca_i$-handling abnormalities associated with the sGC-induced phospholamban phosphorylation.

The activation of the NO-sGC-cGMP pathway also enhances the activity of sodium-calcium exchangers (NCX), contributing to the acceleration of $Ca_i$ homeostasis. A cultured cell model study demonstrated that 8-Br-cGMP, an analog of cGMP, enhances $Na^+$-dependent $Ca^{2+}$ efflux and influx [24], suggesting that the elevation of cGMP stimulates both the forward and reverse modes of NCX. Another study further demonstrated that the NO precursor, L-arginine, increases NCX activity in a cGMP-dependent pathway [25]. The effect of NO-induced NCX upregulation was prevented by ODQ, a guanylate cyclase inhibitor, providing more evidence of the effects of sGC stimulation-related NCX upregulation.

The efficiency of $Ca_i$ removal, or cytosolic $Ca^{2+}$ sequestration, is associated with the development of $Ca^{2+}$-driven alternans. $Ca_i$ removal is mostly dependent on the activity of SERCA and the activity of the NCX. The low efficiency of cytosolic $Ca^{2+}$ sequestration leads to the development of $Ca^{2+}$-driven alternans [26]. Therefore, the overall effects of sGC stimulation on $Ca^{2+}$ include enhanced $Ca^{2+}$ reuptake by SERCA and increased $Ca^{2+}$ efflux/influx by NCX, leading to accelerated $Ca_i$ homeostasis, a shorter $Ca_i$ decay, and suppression of $Ca_i$ alternans.

## Suppression of arrhythmia triggers

The electrophysiological changes seen in ischemic failing hearts encompass several aspects: prolongation of APD, abnormal $Ca_i$ homeostasis, and increased SR $Ca^{2+}$ leak [27]. These factors contribute to an increased susceptibility to ventricular arrhythmias, such as $I_{Ca}$-triggered early afterdepolarization (EAD) and sodium-calcium exchanger (NCX)-induced delayed afterdepolarization (DAD). In this study, we observed a notable reduction in PVB burden following vericiguat therapy. As previously discussed, the activation of sGC appears to play a role in ameliorating abnormal Cai homeostasis. This is achieved through multiple mechanisms: attenuation of the stimulatory effect of beta-adrenergic receptor agonist on $I_{Ca}$, modulation of RYR leak, improvement of $Ca_i$ reuptake by SERCA, and inhibition of $I_{Na}$ [15,20,22]. By diminishing $Ca_i$ levels and reducing $I_{Ca}$ stimulation, the occurrence of early afterdepolarizations (EADs) is suppressed. Moreover, the modulation of RYR leak and $Ca_i$ homeostasis further prevents the activation of NCX and the subsequent development of DADs.

In summary, sGC serves as a pivotal regulator of cardiac ion channels and intracellular calcium homeostasis, operating through the intricate interplay of the NO/sGC/cGMP signaling

pathways. Through these modulatory mechanisms, sGC stimulation may indirectly impact cardiac electrophysiology, thereby contributing to the suppression of cardiac arrhythmias.

## Study limitations

In a pharmacological investigation, the average peak plasma concentration reached 479 μg/L (equivalent to 1.12 μmol/L) in healthy individuals who were administered a single clinical dose (10 mg) of vericiguat therapy [28]. According to a previous study [12], the $EC_{50}$ of vericiguat for sGC activation is 1.005 ± 0.145 mmol/L. Vericiguat at a dose of 1 to 10 μmol/L is considered safe and capable of promoting an increase in cGMP in cardiomyocytes [11]. Therefore, we chose a larger dose to ensure the pharmacological effects in the animal model. However, it's important to note that during the optical mapping and electrophysiological experiments, rabbit hearts were subjected to a relatively higher dosage (5 μmol/L) of vericiguat treatment. Consequently, the inferences drawn from these findings might not be entirely transferable to clinical conditions due to the higher dosage employed in the rabbit model.

Our previous studies demonstrated that the sham hearts were healthy and had extremely low ventricular arrhythmia inducibility [29,30]. Therefore, in some of the disease models, we did not perform sham control animal preparations [8] to reduce animal use, as per the request of the IACUC. Consequently, we did not conduct experiments on sham-operated rabbits in this study.

To mitigate motion-related artifacts, the optical mapping experiments incorporated the use of an excitation-contraction uncoupler called blebbistatin. Notably, blebbistatin carries various inherent cardiac electrophysiological effects, including the prolongation of APD [31]. These modified electrophysiological characteristics induced by blebbistatin infusion could potentially impact the interpretation of optical mapping records.

In this study, we also observed a small APD prolongation of 7~9% with acute vericiguat therapy. Whether the suppression of ventricular arrhythmias is attributed to the small changes of APD is not clear. The contributions of the small changes in APD to the suppression of ventricular arrhythmias need further study.

The rabbits in this study are generally young (~ 6 months old). The above-mentioned results and the interpretations might not be entirely applicable to older rabbits.

## Conclusions

In the MI rabbit model, acute vericiguat therapy demonstrates anti-ventricular arrhythmia effects. These effects can be attributed to various mechanisms, including prolongation of VERP, increase of APD, acceleration of $Ca_i$ homeostasis, and suppression of cardiac alternans. The combined effects result in the prevention of functional conduction block and the avoidance of reentry formation. Moreover, our observations indicate a reduction in the PVB burden, further contributing to the mitigation of ventricular arrhythmia initiation.

## Supporting information

**S1 Fig. Visually and optical mapping images to identify the scars.** The MI region is indeed the area with scar tissue. We could visually define the scar tissue (as shown by the white scar in the left upper panel), identified it with a fluorescent dye (as indicated by the green area in the left lower panel, as healthy tissue emits orange fluorescence), and confirmed it on the optical mapping images (MI area displayed only obscure signals, as shown by the signals noise in the right panel). Since the signals in the MI zone are faint, we did not include the MI in the data analyses.
(PDF)

**S2 Fig. VERP, APD80 and CaD80 at baseline and the delayed phase (N = 7).** A. VERP and the representative ECG traces. B and C. Action potential duration (APD) at pacing cycle length 300 ms and 200 ms, respectively, and the representative AP traces. D and E. CaD at pacing cycle length 300 ms and 200 ms, respectively, and the representative Cai traces. The comparisons of the representative traces were acquired from the same rabbit. APD80, action potential duration measured to 80% repolarization; CaD80, intracellular calcium transient duration measured to 80% repolarization; VERP, ventricular effective refractory period.
(PDF)

**S3 Fig. Alernans maps at baseline and the delayed phase (N = 7).** A. Summarized results of the cardiac alternans at baseline and the delayed phase. B. Representative Vm and Cai traces. The comparisons of the representative traces and maps were acquired from the same rabbit. PCL, pacing cycle length.
(PDF)

**S1 Data.**
(XLSX)

# Acknowledgments

We would like to thank Laboratory Animal Center, Chang Gung Memorial Hospital, Linkou, Taiwan, for the animal husbandry and care.

# Author Contributions

**Conceptualization:** Po-Cheng Chang, Chung-Chuan Chou.

**Data curation:** Po-Cheng Chang, Hui-Ling Lee, Hung-Ta Wo, Hao-Tien Liu, Ming-Shien Wen.

**Formal analysis:** Po-Cheng Chang, Hui-Ling Lee, Hung-Ta Wo, Hao-Tien Liu.

**Funding acquisition:** Po-Cheng Chang, Chung-Chuan Chou.

**Investigation:** Po-Cheng Chang, Hui-Ling Lee, Hung-Ta Wo, Chung-Chuan Chou.

**Methodology:** Po-Cheng Chang, Hui-Ling Lee, Hung-Ta Wo, Chung-Chuan Chou.

**Project administration:** Po-Cheng Chang.

**Resources:** Po-Cheng Chang.

**Supervision:** Ming-Shien Wen, Chung-Chuan Chou.

**Validation:** Po-Cheng Chang, Hui-Ling Lee, Hung-Ta Wo, Hao-Tien Liu, Ming-Shien Wen, Chung-Chuan Chou.

**Visualization:** Po-Cheng Chang, Hui-Ling Lee, Hung-Ta Wo, Hao-Tien Liu, Ming-Shien Wen, Chung-Chuan Chou.

**Writing – original draft:** Po-Cheng Chang.

**Writing – review & editing:** Po-Cheng Chang, Hui-Ling Lee, Hung-Ta Wo, Hao-Tien Liu, Ming-Shien Wen, Chung-Chuan Chou.

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
