## [Decision Letter · Decision Letter 0]

19 Dec 2023

PONE-D-23-34916Vericiguat Suppresses Ventricular Tachyarrhythmias Inducibility in a Rabbit Myocardial Infarction ModelPLOS ONE

Dear Dr. Chou,

Thank you for submitting your manuscript to PLOS ONE. After careful consideration, we feel that it has merit but does not fully meet PLOS ONE’s publication criteria as it currently stands. Therefore, we invite you to submit a revised version of the manuscript that addresses the points raised during the review process.

We look forward to receiving your revised manuscript.

Kind regards,

Elena G. Tolkacheva, PhD

Academic Editor

PLOS ONE

Journal Requirements:

2. To comply with PLOS ONE submissions requirements, in your Methods section, please provide additional information regarding the experiments involving animals and ensure you have included details on (1) methods of sacrifice and (2) efforts to alleviate suffering.

"This study was supported by the Ministry of Science and Technology, Taiwan (110-2314-B-182A-119- to P.C. Chang) and Chang Gung Medical Foundation (CMRPG3L1202 to P.C. Chang)."

Additional Editor Comments:

Please address comments indicated by the Reviewers.

Reviewers' comments:

Reviewer's Responses to Questions

**Comments to the Author**

1. Is the manuscript technically sound, and do the data support the conclusions?

Reviewer #1: Partly

Reviewer #2: Yes

2. Has the statistical analysis been performed appropriately and rigorously? 

Reviewer #1: Yes

Reviewer #2: Yes

3. Have the authors made all data underlying the findings in their manuscript fully available?

Reviewer #1: Yes

Reviewer #2: No

4. Is the manuscript presented in an intelligible fashion and written in standard English?

Reviewer #1: Yes

Reviewer #2: Yes

5. Review Comments to the Author

Reviewer #1: The present study by Chang et al. utilises the optical mapping technique to investigate how the drug vericiguat, which has recently been approved for the use in HFrEF, affects electrical activity and calcium handling in a rabbit model of myocardial infarction. The authors show that this compound reduced the number of premature ventricular beats in this model, as well as causing alterations in conduction velocity, action potential characteristics, and calcium transients. In general, the experiments have been carried out well, however i do have a number of comments/questions.

1) I am a little confused regarding the animals which underwent the repeated EP study, and then underwent the drug challenge. These are your time controls- so this is great that these have been done- but then are these hearts when drug-perfused also included in the analysis? If so this would mean they would have been on the optical mapping set up even longer than what is shown as the time control (as it would have been control-control-drug, instead of control-drug). Is it these animals that were driving the significance? This needs to be described better in the text, and n numbers should be included in the text and figure legends when describing these.

2) Did the Authors attempt to wash out vericiguat on these preperations?

3) I find it a shame that there are no sham operated animals, as it would have been nice to compare the effects in MI vs Sham. Have these experiments been carried out? If so they need to be included, otherwise there needs to be an addition to the limitations that these experiments were not done.

4) On the mapping isochrones, you clearly label the MI region. Was this all scar tissue, and thats how you defined this? Did you ever quantify the area affected by the MI? Was this similar with all animals? This also brings me onto the methods, where it is stated that either one or two branches of the LCA were ligated. What was the determining factor here, if one or two branches were ligated?

5) When the hearts were being optically imaged, was the flow rate controlled? Did the Authors notice any difference in the amount of liquid coming out- or was this measured- this could be interesting to know, to see if any effects are caused by alterations in vessel diameter etc, as opposed to being due to direct effects on the myocardium.

6) How long was the drug perfused for? It states the time controls were 20 minutes, so did you record the drug records after 20 mins of perfusion. Again, this needs to be clarified in the manuscript.

7) The differences seen in CV, for example, are very small, despite there being significance. Would these small differences actually be physiologically relevant? I think there needs to be some discussion of this

8) In all bar charts, i would like to see the individual data points (perhaps even joining up the before and after drug numbers, to see exactly the trends on an individual heart level)

9) Was Echo carried out pre-MI? It would be nice to see these data too- just to show that there is a decrease in EF.

10) n numbers should be included in all the figure legends

11) When examples are shown in figures comparing pre and post drug, are these taken from the same animal. This should be emphasised in the figure legend as well)

12) Please check the units in some of the figures. For example, in Figure 3, the bar chart states that CV is in cm/ms when i believe it should be in cm/s as it is on the isochrones..

13) I believe the figure legend on Supplementary Figure 2 is incorrect, as I believe this is showing the time control data, but in the legend it mentions the drug treatment.

14) Please check for repetition in the manuscript. For example on lines 209-212, there is discussion again about how the model was made, which maybe isnt really needed at this point.

15) Please check grammar throughout the manuscript. For example in the results section of the abstract it states

'accelerated of intracellular calcium' when it should be 'acceleration of intracellular calcium'. Also there should be 'duration' after action potential in this section.

16) I also think there needs to be a number of additional references added- for example on line 59, when stating the fact that vericiguat has been proven to be beneficial in HF patients.

17) I also wondered why the Authors utilised Labview for analysis, when there are a number of free analysis programs available for analysing these data - for example ElectroMap, Rhythm or COSMAS.

18) When discussing the potential mechanisms of the alterations in Calcium transient, i believe there also needs to be a mention of NCX. Although this is discussed later on, when discussing the tau of the transient, this is one of the major players here. Also, im not sure the L-type Ca current should be mentioned here, as this will be having minimal effects on the value of the downstroke of the transient.

19) Where the rabbits male or female or a mix- this needs to be described in the methods.

20) Im guessing the blebbistatin was included in the Tyrodes solution for the imaging part, and this was continuous? This should be a bit clearer in the text i believe.

21) When the affects of NO are discussed on the sodium current, is this the peak sodium current, or the late component. I believe there needs to be some discussion of that, as it may have implications for NCX, especially if its the late component..

22) At the end of the discussion of IKr vs INa, I believe there should be there needs to be an acknowledgement that other ion channels may also play a role (line 279)

23) When discussing ICa, the authors often discuss beta-adrenergic stimulation. As the Authors did not use this method of inducing arrrhythmias in this model, im not really sure these data can be used as an argument? Did you ever try these experiments in the presence of Isoproterenol for example?

24) I know you show the tape measure in Figure 5A, but please put a scale bar on. Also it may be nice to show this without the MI portion faded out, so the reader can see what this looks like- or at least put another photo of this heart without the MI portion greyed out.

Reviewer #2: In their study “Vericiguat suppresses ventricular tachyarrhythmias inducibility in a rabbit myocardial infarction model” Chou et al. investigate the effect of a vericiguat, a stimulator of guanylate cyclase, onto the susceptibility of infarcted isolated rabbit hearts to ventricular arrhythmias using optical mapping. The study is examplary for demonstrating the utility of optical mapping for studying the mechanisms underlying the arrhythmogenesis of compounds and disease in electrophysiological studies. The study is scientifically sound, rigorously executed and includes a detailed discussion putting their findings into perspective. I have only minor concerns regarding some of the methodological approaches and presentation of the results:

- Please use a format with figures and figure captions in place with figure captions next to the figure and the figure close to where referenced in the text. That way I do not have to scroll back and forth from the beginning to the end of the document all the time.

- The study lacks a conflict of interest statement.

- Please add age and sex of rabbits.

- Are there representative APD maps that complement the bar plot in Fig. 2A?

- Please add the number of hearts in Fig. 2 A-E.

- The bar plots do not reveal the number of individual measurements and their spread. Please use a scatter or stripchart plot in addition to bar plots, e.g. as shown here: https://www.ashander.info/notes/barchart-alternatives-in-base-R_files/figure-html/scatter-v-factors-1.png

- Please add an overlay of the traces for one AP or CA-transient so that one can see the effect more clearly or align them in time. The traces as they are presented right now are not helpful for evaluating a change in duration.

- line 186: I feel like the statement “vericiguat therapy significantly slowed the ventricular myocardial conduction” is not supported by the presented data. In the bar plot in Fig. 3A,B the difference is hardly noticeable and I am surprised it is significant based on the p-value (which is a poor measure for statistical significance) given the large error bars. See 2 points below.

- Fig. 3. I do not understand how CV was measured. Please add points from where to where and along which path the distance and CV was measured in the figure.

- Given that the change in CV in the bar plots is not very large: Could this measurement be improved? For instance, the authors could measure the distance between isochrones for every point on the isochrone and compute a CV map and average over the data. Maybe this would produce a more distinct separation between baseline and Vericiquat.

- Fig. 3: Why are there Ca alternates but no V alternates?

- lines 195-198 and Fig. 4B: I would like to see more representative traces that highlight the shortening of the tau value with overlays of the baseline and vericiguat traces.

- Please provide videos for optical mapping data.

- Please provide the video raw data in an appropriate format (e.g. a Matlab or Numpy .mat / .npy file which includes an array) for Fig. 5 D, G.

- line 233: how did the shortening of tau / acceleration of Ca homeostasis contribute to suppression of alternans?

- line 58: missing reference

- line 48: coronary artery disease (CAD) contributes

- lines 318: Why did the authors choose this much higher concentration and what happens when the dosage is reduced, let’s say to 1 micro mol/L?

- line 323: missing reference

- In lines 298-299 the authors mention electrophysiological changes in ischemic failing hearts including slowing of CV and prolongation of AP, which they also report with vericiguat. How can these changes be pro- and anti-arrhythmic at the same time?

6. PLOS authors have the option to publish the peer review history of their article (what does this mean?). If published, this will include your full peer review and any attached files.

Reviewer #1: No

Reviewer #2: No

---

## [Author Response · Author response to Decision Letter 0]

9 Jan 2024

Responses to Reviewers’ Comments

Reviewer #1: The present study by Chang et al. utilises the optical mapping technique to investigate how the drug vericiguat, which has recently been approved for the use in HFrEF, affects electrical activity and calcium handling in a rabbit model of myocardial infarction. The authors show that this compound reduced the number of premature ventricular beats in this model, as well as causing alterations in conduction velocity, action potential characteristics, and calcium transients. In general, the experiments have been carried out well, however i do have a number of comments/questions.

Response: We appreciate the reviewer’s insightful comments after the detailed review. They have greatly helped us improve the manuscript.

1. I am a little confused regarding the animals which underwent the repeated EP study, and then underwent the drug challenge. These are your time controls- so this is great that these have been done- but then are these hearts when drug-perfused also included in the analysis? If so this would mean they would have been on the optical mapping set up even longer than what is shown as the time control (as it would have been control-control-drug, instead of control-drug). Is it these animals that were driving the significance? This needs to be described better in the text, and n numbers should be included in the text and figure legends when describing these.

Response:

We appreciate the reviewer for raising questions to clarify any confusion. The delayed control EP tests involved only a simple procedure, comprising two S1S1 pacing at cycle lengths of 200 ms and 300 ms conducted immediately before vericiguat perfusion. The delayed control tests, lasting less than 10 minutes, were not considered as separate control tests. The purpose of these brief delayed tests was to address concerns related to fluorescence dye decay (photobleaching) and alterations of electrophysiological property associated with the excitation-contraction uncoupler (O’Shea C et al., International Journal of Biochemistry and Cell Biology, 2020; 126: 1058; Efimov IR et al., Circ Res, 2010; 106: 255-71). We randomly selected 7 rabbits out of the 14 for the delayed EP tests. The entire study still adhered to a control-drug protocol. We have revised the text in the Methods section (lines 149-152) and Supplementary Figures. 

2. Did the Authors attempt to wash out vericiguat on these preparations?

Response: We appreciate the reviewer’s suggestion. Unfortunately, conducting a vericiguat washout study in the whole heart optical mapping preparation is challenging. Although the extracellular concentration of the drug can be washed out within a short period, the effects of the intracellular signal pathway take longer to restore a drug-free status. Meanwhile, the fluorescence dyes and the excitation-contraction uncoupler are also partially washed out, leading to a significant alteration of optical images (O’Shea C et al., International Journal of Biochemistry and Cell Biology, 2020; 126: 1058; Efimov IR, et al., Circ Res, 2010; 106: 255-71). After a prolonged washout period, the fluorescence dyes would not function for continuous optical recording. Additional loading of dyes is not useful due to the remaining fluorescent artifacts of photobleached fluorescence. Therefore, a washout study in the whole heart optical mapping preparation is not practical. Consequently, we conducted the aforementioned “delayed” experiments to include a time control and exclude potential effects, rather than relying on a washout test.

3. I find it a shame that there are no sham operated animals, as it would have been nice to compare the effects in MI vs Sham. Have these experiments been carried out? If so they need to be included, otherwise there needs to be an addition to the limitations that these experiments were not done.

Response: Our previous studies demonstrated that the sham hearts were healthy and had extremely low ventricular arrhythmia inducibility (e.g., Chang PC et al., Cardiovasc Ther. 2019: 6032631; Chang PC et al., J Cardiovasc Pharmacol. 2018; 72: 97-105). Therefore, in some of the disease models, we did not perform sham control animal preparations (e.g., Chang PC et al., J Card Fail. 2020; 26: 527-537) to reduce animal use, as per the request of the Institutional Animal Care and Use Committee (IACUC).

We have made revisions to the text in the Limitations section (lines 361-364).

4. On the mapping isochrones, you clearly label the MI region. Was this all scar tissue, and thats how you defined this? Did you ever quantify the area affected by the MI? Was this similar with all animals? This also brings me onto the methods, where it is stated that either one or two branches of the LCA were ligated. What was the determining factor here, if one or two branches were ligated?

Response: The MI region is indeed the area with scar tissue. We visually defined the scar tissue (depicted by the white scar in the left upper panel), identified it with a fluorescent dye (indicated by the green area in the left lower panel, as healthy tissue emits orange fluorescence), and confirmed it in the optical mapping images (the MI area displayed only obscure signals, as shown by the signal noise in the right panel). Since the signals in the MI zone are obscure, we did not include the MI in the data analyses.

Regarding the quantity of scar tissue, we did not have the opportunity to measure it. Vericiguat therapy does not contribute to changes in scar formation. Therefore, we did not consider measuring the scar size using triphenyl tetrazolium chloride (TTC) staining.

We created MI with ligation at the obtuse marginal branches of the left circumflex artery. During coronary ligation, we ensured the development of MI, evident by tissue cyanotic changes over the ligated zone. It is possible that ligation of certain smaller obtuse marginal branches did not induce MI. In some rabbits, a single artery ligation did not cause obvious MI (no cyanotic change) due to collateral circulation. Therefore, we performed an additional artery ligation to create a “true” MI rather than a “sham” ligation.

We have made revisions to the text in the Methods section (lines 89-99).

5. When the hearts were being optically imaged, was the flow rate controlled? Did the Authors notice any difference in the amount of liquid coming out- or was this measured- this could be interesting to know, to see if any effects are caused by alterations in vessel diameter etc, as opposed to being due to direct effects on the myocardium.

Response: The Langendorff perfusion system operated under pressure control. We manually adjusted the pump flow rate to maintain aortic perfusion pressure within the range of 70-90 cmH2O. The flow rate is directly linked to heart rate, contractions, and the health of available vessels and myocardium. Initially, when the isolated hearts were still beating, the perfusion flow rate was approximately 50-60 mL/minute. Following the addition of the excitation-contraction uncoupler (blebbistatin) to mitigate heart contraction, the flow rate was reduced to 20-30 ml/minute.

6. How long was the drug perfused for? It states the time controls were 20 minutes, so did you record the drug records after 20 mins of perfusion. Again, this needs to be clarified in the manuscript.

Response: Thanks for the reminder about the statement on drug perfusion. We initiated the EP tests and optical mapping recording 20 minutes after the start of drug perfusion. We have made revisions to the text in the Methods section (lines 155-157).

7. The differences seen in CV, for example, are very small, despite there being significance. Would these small differences actually be physiologically relevant? I think there needs to be some discussion of this.

Response: In this study, we found that a decrease in conduction velocity (CV) offset the prolongation of action potential duration (APD) in the wavelength of wave propagation. Prolongation of APD increases the wavelength (wavelength = APD x CV, Weiss JN, et al. Circulation. 2005; 112: 1232-1240.), leading to a higher probability of wave break. In our investigation, the decrease in CV alone might offset the electrophysiological effects of APD prolongation regarding the wavelength, preventing the formation of wave breaks during rapid pacing. These changes in CV might protect the rabbit hearts from wave breaks and ventricular tachyarrhythmias. We have revised the text in the Discussions section (lines 297-301).

8. In all bar charts, i would like to see the individual data points (perhaps even joining up the before and after drug numbers, to see exactly the trends on an individual heart level)

Response: We appreciate the reviewer's comment and have revised the bar charts using a connected scatterplot accordingly.

9. Was Echo carried out pre-MI? It would be nice to see these data too- just to show that there is a decrease in EF.

Response: According to our previous data (Chang PC et al., Cardiovasc Ther 2019; 2019: 6032631; Chang PC et al., J Cardiovasc Pharmacol 2018; 72: 97–105), animals without MI creation exhibited normal heart function. We did not routinely perform pre-MI echocardiography in subsequent studies (Chang PC et al., Journal of Cardiac Failure 2020; 26: 527-537). In this study, we conducted cardiac echo to evaluate heart function in only 7 out of 14 rabbits; the left ventricular ejection fraction (LVEF) was 67.6±2.6%, and the post-MI LVEF of these particular rabbits was 37.7±6.2% (P<0.001). This cardiac echo exam was solely for quality control, confirming the presence of normal heart function before surgery.

10. n numbers should be included in all the figure legends

Response: We appreciate the reviewer for the suggestion and have revised the Figures legend according to the comment.

11. When examples are shown in figures comparing pre and post drug, are these taken from the same animal. This should be emphasised in the figure legend as well)

Response: We appreciate the reviewer for the suggestion and have revised the Figures legends according to the comment.

12. Please check the units in some of the figures. For example, in Figure 3, the bar chart states that CV is in cm/ms when i believe it should be in cm/s as it is on the isochrones.

Response: We appreciate the reviewer for the reminder and have revised the Figures according to the comment.

13. I believe the figure legend on Supplementary Figure 2 is incorrect, as I believe this is showing the time control data, but in the legend it mentions the drug treatment.

Response: We appreciate the reviewer for the reminder and have revised the Supplementary Figure 2 legend accordingly.

14. Please check for repetition in the manuscript. For example on lines 209-212, there is discussion again about how the model was made, which maybe isnt really needed at this point.

Response: We appreciate the reviewer for the reminder. In the Results line 209-212, we mentioned a particular example of MI rabbit and the finding of the model creation rather than repetition. 

15. Please check grammar throughout the manuscript. For example in the results section of the abstract it states, accelerated of intracellular calcium' when it should be 'acceleration of intracellular calcium'. Also there should be 'duration' after action potential in this section.

Response: We appreciate the reviewer for the comment and have revised the manuscript accordingly. In some cases, 'action potential' refers to the electrical wave itself, for example, in the context of action potential alternans.

16. I also think there needs to be a number of additional references added- for example on line 59, when stating the fact that vericiguat has been proven to be beneficial in HF patients.

Response: We appreciate the reviewer for the comment and have revised the Introduction accordingly. 

17. I also wondered why the Authors utilised Labview for analysis, when there are a number of free analysis programs available for analysing these data - for example ElectroMap, Rhythm or COSMAS.

Response: Our lab has utilized LabVIEW along with an add-on app, MiCAM Viewer, courtesy of Professor Shien-Fong Lin, for approximately 20 years. He has developed several customized functions for optical mapping analyses. However, we are interested in exploring new, free analysis programs. Thank you for your comment.

18. When discussing the potential mechanisms of the alterations in Calcium transient, i believe there also needs to be a mention of NCX. Although this is discussed later on, when discussing the tau of the transient, this is one of the major players here. Also, im not sure the L-type Ca current should be mentioned here, as this will be having minimal effects on the value of the downstroke of the transient.

Response: We appreciate the reviewer for the comment and have revised the Discussions accordingly (lines 319-332). 

19. Where the rabbits male or female or a mix- this needs to be described in the methods.

Response: In this study, we used both female and male rabbits, and we have accordingly revised the Methods section (lines 168-169). 

20. Im guessing the blebbistatin was included in the Tyrodes solution for the imaging part, and this was continuous? This should be a bit clearer in the text i believe.

Response: That is correct. We used blebbistatin in the Tyrode’s solution continuously during the imaging acquisition. Thank you for the suggestion. We have revised the text in the Methods section accordingly (Lines 113-115).

21. When the affects of NO are discussed on the sodium current, is this the peak sodium current, or the late component. I believe there needs to be some discussion of that, as it may have implications for NCX, especially if its the late component.

Response: We appreciate the reviewer for the comment and have revised the Discussions accordingly(lines 319-332).

22. At the end of the discussion of IKr vs INa, I believe there should be there needs to be an acknowledgement that other ion channels may also play a role (line 279).

Response: We appreciate the reviewer for the comment and have revised the Discussions accordingly (lines 285-296).

23. When discussing ICa, the authors often discuss beta-adrenergic stimulation. As the Authors did not use this method of inducing arrrhythmias in this model, im not really sure these data can be used as an argument? Did you ever try these experiments in the presence of Isoproterenol for example?

Response: We appreciate the reviewer's comment. According to the guidelines, typical protocols for electrophysiological studies include stimulation from 2 ventricular sites with 2-3 basic drive cycle lengths, induction of 3 extra-stimuli, and isoproterenol administration. However, the yield of programmed electrical stimulation varies with the underlying cardiac condition and its severity, the presence or absence of spontaneous VT, concomitant drug therapy, stimulation protocol, and site(s) of stimulation (2023 ESC guidelines for the management of patients with ventricular arrhythmias and the prevention of sudden cardiac death, European Heart Journal 2022; 43: 3997–4126). For a model with MI and reduced ejection fraction, the yield of electrical pacing-induced ventricular arrhythmias is relatively higher. Therefore, we did not use additional isoproterenol administration.

24. I know you show the tape measure in Figure 5A, but please put a scale bar on. Also it may be nice to show this without the MI portion faded out, so the reader can see what this looks like- or at least put another photo of this heart without the MI portion greyed out.

Response: In Figure 5A, a ruler is present on the rabbit heart, and we have removed the MI portion label. Thanks for the recommendation. 

Reviewer #2: In their study “Vericiguat suppresses ventricular tachyarrhythmias inducibility in a rabbit myocardial infarction model” Chou et al. investigate the effect of a vericiguat, a stimulator of guanylate cyclase, onto the susceptibility of infarcted isolated rabbit hearts to ventricular arrhythmias using optical mapping. The study is examplary for demonstrating the utility of optical mapping for studying the mechanisms underlying the arrhythmogenesis of compounds and disease in electrophysiological studies. The study is scientifically sound, rigorously executed and includes a detailed discussion putting their findings into perspective. I have only minor concerns regarding some of the methodological approaches and presentation of the results:

Response: We appreciate the reviewer’s insightful comments after the detailed review. They have greatly helped us improve the manuscript.

1. Please use a format with figures and figure captions in place with figure captions next to the figure and the figure close to where referenced in the text. That way I do not have to scroll back and forth from the beginning to the end of the document all the time.

Response: We appreciate the reviewer for the comment and made an additional version to make the reviewing process more comfortable.

2. The study lacks a conflict of interest statement.

Response: We appreciate the reviewer for the comment and have add conflict of interest statement accordingly (both ICMJE Disclosure Form and a statement in the manuscript).

3. Please add age and sex of rabbits.

Response: We appreciate the reviewer's advice. The average age of rabbits during the optical mapping study was 25.1 ± 2.5 weeks. For this study, we included a total of 7 female and 7 male rabbits in the optical mapping analyses. Accordingly, we have revised the Results sections (lines 168-171).

4. Are there representative APD maps that complement the bar plot in Fig. 2A?

Response: We appreciate the reviewer’s suggestion and have incorporated APD and Cai duration maps into Figure 2. 

5. Please add the number of hearts in Fig. 2 A-E.

Response: We have added heart number to Figure 2 accordingly. 

6. The bar plots do not reveal the number of individual measurements and their spread. Please use a scatter or stripchart plot in addition to bar plots, e.g. as shown here: https://www.ashander.info/notes/barchart-alternatives-in-base-R_files/figure-html/scatter-v-factors-1.png

Response: We appreciate the reviewer's suggestion and have revised the bar charts using a connected scatterplot, as per the comment.

7. Please add an overlay of the traces for one AP or CA-transient so that one can see the effect more clearly or align them in time. The traces as they are presented right now are not helpful for evaluating a change in duration.

Response: We appreciate the reviewer's suggestion and have replaced the original AP and Cai traces with overlaid traces.

8. line 186: I feel like the statement “vericiguat therapy significantly slowed the ventricular myocardial conduction” is not supported by the presented data. In the bar plot in Fig. 3A,B the difference is hardly noticeable and I am surprised it is significant based on the p-value (which is a poor measure for statistical significance) given the large error bars. See 2 points below.

Response: We appreciate the reviewer's comment. Due to the variable conduction velocity among different hearts, the variation was significant. Nevertheless, the paired comparisons still demonstrated statistically significant differences between the baseline and vericiguat therapy. We have replaced the original bar graphs with connected scatterplots.

9. Fig. 3. I do not understand how CV was measured. Please add points from where to where and along which path the distance and CV was measured in the figure.

Response: We appreciate the reviewer's comment. In response, we added points to illustrate the path and the locations where conduction velocity (CV) was measured. Taking Figure 3A baseline as an example, the pacing site is located on the left (indicated by the pacing site symbol), and the arrow indicates the conduction path. We measured the distance between the arrow tail and the arrowhead (1.56 cm in this heart) and the conduction duration (22 ms). The calculated conduction velocity was 70.80 cm/s.

10. Given that the change in CV in the bar plots is not very large: Could this measurement be improved? For instance, the authors could measure the distance between isochrones for every point on the isochrone and compute a CV map and average over the data. Maybe this would produce a more distinct separation between baseline and Vericiquat.

Response: We appreciate the reviewer's comment. The issue is not a measurement problem. Due to the variable conduction velocity among different hearts, the variation was significant.

11. Fig. 3: Why are there Ca alternates but no V alternates?

Response: We thank the reviewer for raising the question. Membrane potential (Vm) alternans manifested as changes in action potential duration (APD) from short to long, while calcium (Cai) alternans presented as amplitude changes from low to high. We observed both Vm and Cai alternans.

12. lines 195-198 and Fig. 4B: I would like to see more representative traces that highlight the shortening of the tau value with overlays of the baseline and vericiguat traces.

Response: We appreciate the reviewer's comment. Because the Cai decay depends on the Cai curve and its fitting. Additional overlaid traces of the baseline and vericiguat might not be good to show the changes (for example the below traces).

13. Please provide videos for optical mapping data.

Response: We appreciate the reviewer's comment. We included the representative videos in the supplementary data.

Please refer to 

https://1drv.ms/f/s!AnJU7ssgUKgtg8FXi8YU7EWNbhZ1Fw?e=SQeDSm

14. Please provide the video raw data in an appropriate format (e.g. a Matlab or Numpy .mat / .npy file which includes an array) for Fig. 5 D, G.

Response: The raw data for MiCAM images is in “rsd” format. Unfortunately, we were unable to transform the format into either “mat” or “npy.” We have uploaded the raw data in “rsd” format, zipped into a file (“Pacing induced VT raw data for Fig 5.zip”). 

Please refer to 

https://1drv.ms/f/s!AnJU7ssgUKgtg8FXi8YU7EWNbhZ1Fw?e=SQeDSm

15. line 233: how did the shortening of tau / acceleration of Ca homeostasis contribute to suppression of alternans?

Response: We appreciate the reviewer's question. The efficiency of Cai removal, or cytosolic Ca2+ sequestration, is associated with the development of Ca2+-driven alternans. Cai removal is mostly dependent on the activity of SERCA and the activity of the NCX. Low efficiency of cytosolic Ca2+ sequestration leads to the development of Ca2+-driven alternans. Therefore, the overall effects of sGC stimulation on Ca2+ include enhanced Ca2+ reuptake by SERCA and increased Ca2+ efflux/influx by NCX, leading to accelerated Cai homeostasis, a shorter Cai decay, and suppression of Cai alternans. We have added some descriptions in the Discussions section.

16. line 58: missing reference

Response: We thank the reviewer for the reminder.

17. line 48: coronary artery disease (CAD) contributes

Response: We thank the reviewer for the reminder. We have revised the text accordingly.

18. lines 318: Why did the authors choose this much higher concentration and what happens when the dosage is reduced, let’s say to 1 micro mol/L?

Response: According to previous publications (J Med Chem 60, 5146-5161; Am J Health Syst Pharm 78, 1021-1023), the EC50 of vericiguat for sGC activation is 1.005 ± 0.145 mmol/L. As reported by Cai et al. (Ann Transl Med. 2022; 10: 662.), vericiguat at a dose of 1 to 10 µmol/L is considered safe and capable of promoting an increase in cGMP in cardiomyocytes. Therefore, we chose a larger dose to ensure the pharmacological effects in the animal model.

19. line 323: missing reference

Response: We thank the reviewer for the reminder.

20. In lines 298-299 the authors mention electrophysiological changes in ischemic failing hearts including slowing of CV and prolongation of AP, which they also report with vericiguat. How can these changes be pro- and anti-arrhythmic at the same time?

Response: We appreciate the reviewer’s comment. Many anti-arrhythmic agents exhibit both anti- and pro-arrhythmic properties simultaneously. The overall composite effects determine whether an agent acts as an anti- or pro-arrhythmic agent for a specific arrhythmia (e.g., ventricular tachycardia) in a particular context (e.g., ischemic heart). In this MI model, we believe the composite effects of vericiguat contribute to its anti-arrhythmic properties. These reasons include the enhancement of conduction, suppression of cardiac alternans, and inhibition of arrhythmic triggers. The collective effects render vericiguat anti-arrhythmic.

On the other hand, prolongation of APD increases the wavelength (wavelength = APD × CV), leading to a higher probability of wave break (Weiss JN, et al. Circulation. 2005; 112: 1232-1240). In this study, the decrease in CV after vericiguat therapy might offset the pro-arrhythmic effects of APD prolongation regarding the wavelength, preventing the formation of wave breaks during rapid pacing. The decrease of CV might protect the rabbit hearts from wave breaks and the development of ventricular tachyarrhythmias.

We have made revision in the Discussions section (lines 297-301).

---

## [Decision Letter · Decision Letter 1]

14 Feb 2024

PONE-D-23-34916R1Vericiguat Suppresses Ventricular Tachyarrhythmias Inducibility in a Rabbit Myocardial Infarction ModelPLOS ONE

Dear Dr. Chou,

Thank you for submitting your manuscript to PLOS ONE. After careful consideration, we feel that it has merit but does not fully meet PLOS ONE’s publication criteria as it currently stands. Therefore, we invite you to submit a revised version of the manuscript that addresses the points raised during the review process. Please submit your revised manuscript by Mar 30 2024 11:59PM. If you will need more time than this to complete your revisions, please reply to this message or contact the journal office at plosone@plos.org. Please include the following items when submitting your revised manuscript:A rebuttal letter that responds to each point raised by the academic editor and reviewer(s). You should upload this letter as a separate file labeled 'Response to Reviewers'.A marked-up copy of your manuscript that highlights changes made to the original version. You should upload this as a separate file labeled 'Revised Manuscript with Track Changes'.An unmarked version of your revised paper without tracked changes. You should upload this as a separate file labeled 'Manuscript'.If applicable, we recommend that you deposit your laboratory protocols in protocols.io to enhance the reproducibility of your results. Protocols.io assigns your protocol its own identifier (DOI) so that it can be cited independently in the future. For instructions see: https://journals.plos.org/plosone/s/submission-guidelines#loc-laboratory-protocols. Additionally, PLOS ONE offers an option for publishing peer-reviewed Lab Protocol articles, which describe protocols hosted on protocols.io. Read more information on sharing protocols at https://plos.org/protocols?utm_medium=editorial-email&utm_source=authorletters&utm_campaign=protocols.

We look forward to receiving your revised manuscript.

Kind regards,

Elena G. Tolkacheva, PhD

Academic Editor

PLOS ONE

Journal Requirements:

**Additional Editor Comments:**

Please address minor comments indicated by the reviewer.

Reviewers' comments:

Reviewer's Responses to Questions

**Comments to the Author**

1. If the authors have adequately addressed your comments raised in a previous round of review and you feel that this manuscript is now acceptable for publication, you may indicate that here to bypass the “Comments to the Author” section, enter your conflict of interest statement in the “Confidential to Editor” section, and submit your "Accept" recommendation.

Reviewer #1: All comments have been addressed

Reviewer #2: (No Response)

2. Is the manuscript technically sound, and do the data support the conclusions?

Reviewer #1: Yes

Reviewer #2: Yes

3. Has the statistical analysis been performed appropriately and rigorously? 

Reviewer #1: Yes

Reviewer #2: I Don't Know

4. Have the authors made all data underlying the findings in their manuscript fully available?

Reviewer #1: Yes

Reviewer #2: No

5. Is the manuscript presented in an intelligible fashion and written in standard English?

Reviewer #1: Yes

Reviewer #2: Yes

6. Review Comments to the Author

Reviewer #1: All comments have been addressed which i thank the Authors for. Please check grammar and spelling throughout the manuscript though- for instance on line 272, cGC is stated instead of sGC..

Reviewer #2: Thank you very much for the detailed response and revision of the manuscript. I have a few remaining minor concerns, the main concern being that the measured effects (APD / CV) are very small and I would like to verify the effect in the raw data myself (PlosONE specifically asks me whether all raw data was made available).

- Please mark all revised parts of your manuscript in blue, otherwise it is difficult to confirm which parts were revised.

- Thank you very much for providing raw data (Pacing induced VT raw data for Fig 5). Could you please provide raw data for Figs. 2 and 3 which provides a comparison of baseline and Vericiguat? Please provide several paired corresponding videos with baseline and Vericiguat so that I can reproduce and verify the plots in Figs. 2 and 3.

- The authors should phrase the significance of the measured APD and CV changes more carefully. They may be statistically significant, but still small.

- The rabbits are very young (6 months). The limitations section should list this as a potential limitation (would the results change with older rabbits?).

As a note:

You can use the following software for the post-processing of optical mapping videos: https://github.com/cardiacvision/optimap

Tutorial 13 explains how to read in .rsh videos: https://optimap.readthedocs.io/en/latest/tutorials/io/

7. PLOS authors have the option to publish the peer review history of their article (what does this mean?). If published, this will include your full peer review and any attached files.

Reviewer #1: No

Reviewer #2: No

---

## [Author Response · Author response to Decision Letter 1]

28 Feb 2024

Responses to reviewers’ comments

Reviewer #1: All comments have been addressed which i thank the Authors for. Please check grammar and spelling throughout the manuscript though- for instance on line 272, cGC is stated instead of sGC.

Response: We appreciate the reviewer’s comments and effort. We have revised the manuscript accordingly. 

 

Reviewer #2: Thank you very much for the detailed response and revision of the manuscript. I have a few remaining minor concerns, the main concern being that the measured effects (APD / CV) are very small and I would like to verify the effect in the raw data myself (PlosONE specifically asks me whether all raw data was made available).

Response: We appreciate the reviewer’s comments and effort. We have uploaded the videos, as well as the optical mapping raw data, as suggested. 

- Please mark all revised parts of your manuscript in blue, otherwise it is difficult to confirm which parts were revised.

Response: Thank you for the suggestion. Previously, we uploaded a file with yellow-highlighted texts to indicate the revisions made in the R1 version. In this R2 revision, we have marked the revised texts in blue (including the revised texts in R1).

- Thank you very much for providing raw data (Pacing induced VT raw data for Fig 5). Could you please provide raw data for Figs. 2 and 3 which provides a comparison of baseline and Vericiguat? Please provide several paired corresponding videos with baseline and Vericiguat so that I can reproduce and verify the plots in Figs. 2 and 3.

Response: We appreciate the reviewer’s comments. In the R2 revision, we have uploaded several corresponding videos and raw data at baseline and with vericiguat therapy. Per the request from editorial office, we have removed the figures and legends from the main text. We apologize for the inconvenience.

Please refer to 

Videos: 

Videos for Fig 2 and 3 

https://1drv.ms/f/s!AnJU7ssgUKgtg8QdlGG-95PAVomUKw?e=to10Pn

Optical mapping raw data:

OM raw data 

https://1drv.ms/f/s!AnJU7ssgUKgtg8Ia4nMt8oC9wxZGOw?e=DjEexX

- The authors should phrase the significance of the measured APD and CV changes more carefully. They may be statistically significant, but still small.

Response: We appreciate the reviewer’s suggestions. We recognize the small changes in APD and CV. However, the significant small CV reduction and APD prolongation might contribute, at least partly, to the observed electrophysiological effects of vericiguat in the chronic MI model. Although the APD and CV changes were small, the combination of the changes as well as Ca homeostasis led to suppression of discordant alternans and functional conduction block/ wave break. Thereby, the ventricular arrhythmias inducibility was also suppressed. We have revised the text to mention the small changes in APD and CV in the Discussions (line 236; lines 364-370).

- The rabbits are very young (6 months). The limitations section should list this as a potential limitation (would the results change with older rabbits?).

Response: Certainly, the rabbits are relatively young. We appreciate the reviewer for the comment and concern. We have revised the limitations (lines 371-372).

---

## [Decision Letter · Decision Letter 2]

12 Mar 2024

PONE-D-23-34916R2Vericiguat Suppresses Ventricular Tachyarrhythmias Inducibility in a Rabbit Myocardial Infarction ModelPLOS ONE

Dear Dr. Chou,

Thank you for submitting your manuscript to PLOS ONE. After careful consideration, we feel that it has merit but does not fully meet PLOS ONE’s publication criteria as it currently stands. Therefore, we invite you to submit a revised version of the manuscript that addresses the points raised during the review process.

Please address comments indicated by the Reviewer.

We look forward to receiving your revised manuscript.

Kind regards,

Elena G. Tolkacheva, PhD

Academic Editor

PLOS ONE

Journal Requirements:

Reviewers' comments:

Reviewer's Responses to Questions

**Comments to the Author**

1. If the authors have adequately addressed your comments raised in a previous round of review and you feel that this manuscript is now acceptable for publication, you may indicate that here to bypass the “Comments to the Author” section, enter your conflict of interest statement in the “Confidential to Editor” section, and submit your "Accept" recommendation.

Reviewer #2: (No Response)

2. Is the manuscript technically sound, and do the data support the conclusions?

Reviewer #2: Partly

3. Has the statistical analysis been performed appropriately and rigorously? 

Reviewer #2: No

4. Have the authors made all data underlying the findings in their manuscript fully available?

Reviewer #2: No

5. Is the manuscript presented in an intelligible fashion and written in standard English?

Reviewer #2: Yes

6. Review Comments to the Author

Reviewer #2: Thank you very much for providing the requested raw data. I confirm that the APD prolongation occurs with Vericiguat in all data I reviewed. However, I cannot confirm the findings regarding the change in CV. I attached 2 videos showing a comparison of the wave propagation at Baseline vs. Vericiguat (each in rabbits 1 and 2, recordings 200.rsh). In rabbit 1, I do not notice any difference in the wave speed and in rabbit 2 the wave activates the ventricle with Vericiguat first, even though the wave comes in later, suggesting transmural conduction or a much faster CV with Vericiguat. In any way, the data I reviewed does not support that Vericiguat decreases CV, and I therefore suggest removing the data and claim or revising the paper and presenting the analysis in a convincing fashion. I leave this to the authors and recommend' accept' if the CV data is removed.

7. PLOS authors have the option to publish the peer review history of their article (what does this mean?). If published, this will include your full peer review and any attached files.

Reviewer #2: No

---

## [Author Response · Author response to Decision Letter 2]

22 Mar 2024

Responses to reviewers’ comments

Reviewer #2: Reviewer #2: Thank you very much for providing the requested raw data. I confirm that the APD prolongation occurs with Vericiguat in all data I reviewed. However, I cannot confirm the findings regarding the change in CV. I attached 2 videos showing a comparison of the wave propagation at Baseline vs. Vericiguat (each in rabbits 1 and 2, recordings 200.rsh). In rabbit 1, I do not notice any difference in the wave speed and in rabbit 2 the wave activates the ventricle with Vericiguat first, even though the wave comes in later, suggesting transmural conduction or a much faster CV with Vericiguat. In any way, the data I reviewed does not support that Vericiguat decreases CV, and I therefore suggest removing the data and claim or revising the paper and presenting the analysis in a convincing fashion. I leave this to the authors and recommend' accept' if the CV data is removed.

Response: We appreciate the reviewer’s comments and effort. The data in the Results and the associated texts in the abstract and Discussions have been removed. We have revised the manuscript accordingly.

---

## [Decision Letter · Decision Letter 3]

27 Mar 2024

Vericiguat Suppresses Ventricular Tachyarrhythmias Inducibility in a Rabbit Myocardial Infarction Model

PONE-D-23-34916R3

Dear Dr. Chou,

We’re pleased to inform you that your manuscript has been judged scientifically suitable for publication and will be formally accepted for publication once it meets all outstanding technical requirements.

Kind regards,

Elena G. Tolkacheva, PhD

Academic Editor

PLOS ONE

Additional Editor Comments (optional):

Reviewers' comments:

Reviewer's Responses to Questions

**Comments to the Author**

1. If the authors have adequately addressed your comments raised in a previous round of review and you feel that this manuscript is now acceptable for publication, you may indicate that here to bypass the “Comments to the Author” section, enter your conflict of interest statement in the “Confidential to Editor” section, and submit your "Accept" recommendation.

Reviewer #2: All comments have been addressed

2. Is the manuscript technically sound, and do the data support the conclusions?

Reviewer #2: Yes

3. Has the statistical analysis been performed appropriately and rigorously? 

Reviewer #2: I Don't Know

4. Have the authors made all data underlying the findings in their manuscript fully available?

Reviewer #2: No

5. Is the manuscript presented in an intelligible fashion and written in standard English?

Reviewer #2: Yes

6. Review Comments to the Author

Reviewer #2: Thank you very much for removing the part about CV. Please check the PLOS data policy and upload the supporting raw data to a repository before publication.

7. PLOS authors have the option to publish the peer review history of their article (what does this mean?). If published, this will include your full peer review and any attached files.

Reviewer #2: No

---

## [Editor Report · Acceptance letter]

3 Apr 2024

PONE-D-23-34916R3 

PLOS ONE

Dear Dr. Chou, 

I'm pleased to inform you that your manuscript has been deemed suitable for publication in PLOS ONE. Congratulations! Your manuscript is now being handed over to our production team.

Kind regards, 

on behalf of

Dr. Elena G. Tolkacheva 

Academic Editor

PLOS ONE